# Discrete VQ-IHDM: MRI Generation with Vector Quantized Inverse Heat Dissipation Model

## Abstract

Accurate and efficient MRI generation is critical in various clinical settings, such as neurology and radiology. The complex data collection procedures, privacy concerns, and lack of medical experts present a bottleneck in the medical imaging data collection and annotation process. In this paper, we adopt a method to unconditionally generate 2D axial brain MRI using a combination of Vector-Quantized image representation and Inverse Heat Dissipation Model (IHDM). We utilize Gaussian Blur as an alternative to order-agnostic masking in the forward process and train a Transformer model to learn the reverse process. This approach allows us to create a single-step sampling algorithm while maintaining high image fidelity. On the ADNI dataset, our model has a FID score of 38.57, a KID score of 0.036, and an ISC score of 1.84.

## 1 Introduction

Diffusion models have impacted the medical field significantly (Kazerouni et al., 2022) in Image-to-Image translation (Lyu & Wang, 2022; Özbey et al., 2022), Reconstruction (Song et al., 2021; Peng et al., 2022), and Image Generation (Moghadam et al., 2023; Pinaya et al., 2022). Diffusion models (Sohl-Dickstein et al., 2015) are a powerful class of probabilistic generative models that are used to learn complex data distributions. The forward diffusion process gradually adds noise to the data until it transforms to pure Gaussian noise while the reverse diffusion process recovers the structure of the data from the perturbed noise. Additionally, diffusion models can achieve superior image quality compared to GAN-based models (Dhariwal & Nichol, 2021). However, diffusion models tend to have high computational requirements and slow sampling speed. Alternative approaches such as latent diffusion models (Rombach et al., 2021), vector-quantized models (Bond-Taylor et al., 2022) have been developed to counter the high computational requirement of training diffusion models. Conventionally, in the forward process of diffusion models Gaussian noise is added to the samples (Sohl-Dickstein et al., 2015), without taking the structure of data into consideration. Rissanen et al. (2022) show that by iteratively inverting the heat equation, it is possible to generate images with the scope of disentanglement of color and shapes in images.

In this paper, we adopt a hybrid generative model called Discrete VQ-IHDM that is a combination of the VQGAN model from Bond-Taylor et al. (2022) and the forward process of Rissanen et al. (2022) which is known as inverse heat dissipation model or IHDM. Initially, we train our VQGAN model to obtain vector-quantized latent codes. Afterward, we train a latent-diffusion-like model using the Gaussian blur as the forward process from Rissanen et al. (2022). The sampler uses the diffusion model to generate 2D axial brain MRIs with high fidelity and variety. We hope that our model will enable us to create a synthetic dataset for education, learning algorithms, and a dataset for rare subtype diseases.

## 2 Related Works

### 2.1 Diffusion Models

Diffusion models are a class of generative models that define a Markov chain $q(x_{1:T}|x_0) = \prod_{t=1}^{T} q(x_t|x_{t-1})$ of diffusion steps that slowly adds random noise to the data and then learn the reverse process $p_\theta(x_{0:T}) =$

$p_\theta(x_T)\prod_{t=1}^{T}p_\theta(x_{t-1}|x_t)$ to generate desired samples from the noise (Sohl-Dickstein et al., 2015). Over the last couple of years, diffusion models have seen an improvement in model performances due to ideas on training on different levels of resolution and cascading the diffusion models together (Dhariwal & Nichol, 2021; Ho et al., 2021; Saharia et al., 2021). Moreover, diffusion models have been used in conjunction with other models to overcome high computational requirements and produce high-resolution imagery. In the unleashing transformer paper (Bond-Taylor et al., 2022) the authors combined vector-quantized image models to learn an information-rich codebook (Van Den Oord et al., 2017) with discrete diffusion models (Austin et al., 2021; Hoogeboom et al., 2021) which enabled the unconditional generation of globally consistent high-resolution images at of fraction of computational expenses required for diffusion models. To learn the distribution of the information-rich codebook, absorbing state diffusion (Austin et al., 2021) was used where a Transformer (Vaswani et al., 2017) model learned the representations of the information-rich codebook.

Some diffusion models have been developed to consider the inductive biases of the images. In Rissanen et al. (2022) a new model was developed that iteratively generates images by inverting the heat equation. This allows the diffusion model to show emergent qualitative properties such as disentanglement of overall color and shape in images. In this work, the conventional forward step of the diffusion model which is to add noise at each time step (Sohl-Dickstein et al., 2015) has been replaced with a partial differential equation that describes heat dissipation.

$$\frac{\partial}{\partial t}u(x,y,t) = \Delta u(x,y,t) \tag{1}$$

where $u : \mathbb{R}^2 \times \mathbb{R}_+ \to \mathbb{R}$ is the continuous 2D plane of one channel of the image, and $\Delta = \nabla^2$ is the Laplace operator. In this paper, the PDE equation was solved using the eigenbasis of the Laplace operator. Since Neumann boundary conditions are used, the eigenbasis is a cosine basis. As a result, the Laplace operator can be written in terms of finite eigendecomposition $\Delta \triangleq \mathbf{V}\Lambda\mathbf{V}^T$, where $\mathbf{V}^T$ is the cosine basis projection matrix, and $\Lambda$ is a diagonal matrix with negative squared frequencies on the diagonal.

The initial state is projected onto the basis with discrete cosine transform ($\tilde{u} = \mathbf{V}^T\mathbf{u} = \text{DCT}(\mathbf{u})$) in $\mathcal{O}(N\log N)$ time. The solution is given by the finite-dimensional evolution model, describing the decay of frequencies.

$$u(t) = F(t)u(0) = \exp(V\Lambda V^T t)u(0) = V\exp(\Lambda t)V^T u(0) \iff \tilde{u}(t) = \exp(\Lambda t)\tilde{u}(0) \tag{2}$$

where $F(t) \in \mathbb{R}^{N \times N}$ is the transition model and $\mathbf{u}(0)$ is the initial state. The diagonal terms of $\Lambda$ are the negative squared frequencies $-\lambda_{n,m} = -\pi(\frac{n^2}{W^2} + \frac{m^2}{H^2})$, where $W$ and $H$ are the width and height of the image in the pixels $n = 0,....,W-1$ and $m = 0,....,H-1$. Since $\Lambda$ is diagonal, the solution is fast to evaluate and implement with a few lines of code (a code snippet is provided below).

```
import numpy as np
from scipy.fftpack import dct, idct
def heat_eq_forward(u, t):
    # Assuming the image u is a (K x K) numpy array
    K = u.shape[-1]
    freqs = np.pi*np.linspace(0,K-1,K)/K
    frequencies_squared = freqs[:,None]**2 + freqs[None,:]**2
    u_proj = dct(u, axis=0, norm='ortho')
    u_proj = dct(u_proj, axis=1, norm='ortho')
    u_proj = np.exp( - frequencies_squared * t) * u_proj
    u_reconstucted = idct(u_proj, axis=0, norm='ortho')
    u_reconstucted = idct(u_reconstucted, axis=1, norm='ortho')
    return u_reconstucted
```

### 2.2 Diffusion Models in Medical Image Generation

Diffusion models have extensive applications in the medical domain such as image-to-image translation, reconstruction, registration, segmentation, denoising, 2/3D generation, anomaly detection, and other medically-related challenges (Kazerouni et al., 2022). In this section, we are going to cover diffusion model applications in 2/3D generation along with their practical use cases.

In Pinaya et al. (2022) Latent Diffusion Models (LDMs) were trained on T1w MRI images from the UK Biobank dataset (Sudlow et al., 2015). The models were trained to learn the probabilistic distribution of brain images, conditioned on covariables such as age, sex, and brain structure volume. LDMs were used which combined autoencoders to compress the data into a lower-dimensional representation with diffusion models. The autoencoder was trained with a combination of L1 loss, perceptual loss (Zhang et al., 2018), and a patch-based adversarial objective (Esser et al., 2021). After training the compression model, a diffusion model was used to train on the latent (Sohl-Dickstein et al., 2015; Song et al., 2020). A hybrid approach was used by combining the conditions with the input data and using cross-attention mechanisms, for conditioning on covariables based on the approach in Rombach et al. (2021). This approach is able to create realistic data and capable of generating data based on conditioning variables. A synthetic dataset with 100,000 brain images was released along with conditioning information and the link to the dataset can be found in the paper (Pinaya et al., 2022).

Diffusion models have played an important role in the synthesis of histopathology images (Moghadam et al., 2023). Microscopic study of diseased tissue is crucial for cancer diagnosis and prognostication. The generated synthetic histopathology images can be utilized in education, proficiency testing, privacy, and data sharing. In Moghadam et al. (2023), they use DDPMs Ho et al. (2020) with genotype guidance to synthesize images with morphological and genomic information. A color normalization module (Vahadane et al., 2016) was fed with input images to unify the domain of all images which enforces the model to place emphasis on morphological patterns and tackle data consistency problems. Additionally, a morphology levels prioritization module (Choi et al., 2022) was adopted that designates higher weight values to earlier level losses emphasizing perceptual information, resulting in higher fidelity samples. Experiments on the Cancer Genome Atlas (TCGA)(Grossman et al., 2016) dataset exhibit superior performance in contrast to GAN-based approaches (Karras et al., 2017).

One of the earliest works on 3D MRI generation using diffusion models is from (Dorjsembe et al., 2022). They have created a 3D-DDPM by using Ho et al. (2020) as the base model and adopting the cosine noise schedule from Nichol & Dhariwal (2021). All 2D operations, layers, and noise inputs were replaced by their 3D counterparts. The model was trained on the National Taiwan University Hospital's Intracranial Tumor Segmentation (ICTS) dataset (Lu et al., 2021) which contains 1500 contrast-enhanced anonymized T1 images with only 250 steps. Their method outperforms CCE-GAN (Xing et al., 2021) and 3D-$\alpha$-GAN (Kwon et al., 2019) in terms of capturing real data distribution and generating diverse samples.

## 3 Discrete VQ-IHDM

We describe our 2-stage approach for generating 2D axial brain MRI using an Inverse Heat Dissipation Model to represent Vector-Quantized images. In the first stage, we train a VQGAN model to learn the discrete code representation of the images. In the second stage, we use IHDM to train a discrete diffusion model on the discrete code representations. For sampling, we use our discrete diffusion model to generate the discrete code representations which are passed to the generator of VQGAN. Our method is shown in figure 1

### 3.1 Discrete Code

In this stage, we train a VQGAN model to capture the discrete code representations (Van Den Oord et al., 2017) by adopting the hybrid generative model from Bond-Taylor et al. (2022). Given an image our encoder downsamples image $x$ to a smaller spatial resolution, $E(x) = e_1, e_2, ..., e_L \in \mathbb{R}^{L \times D}$. Similar to (Bond-Taylor et al., 2022) as a quantization approach an argmax operation maps the encodings to the closest elements in the codebook of vectors (Van Den Oord et al., 2017). Given a codebook $C \in \mathbb{R}^{K \times D}$, where $K$ is the

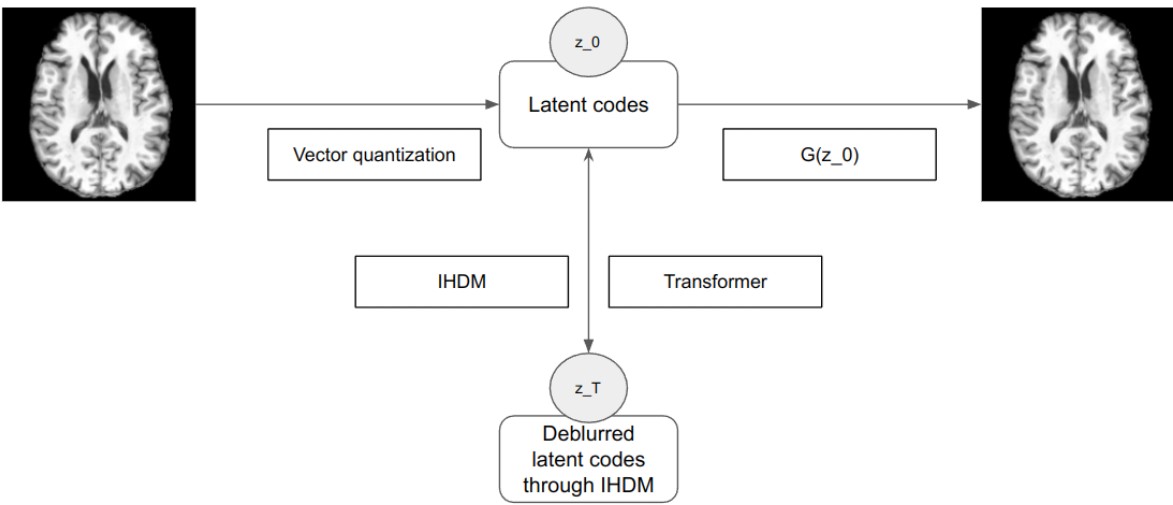

Figure 1: Discrete VQ-IHDM

number of discrete codes in the codebook and $D$ is the dimension of each code, each encoding is mapped via a nearest neighbor lookup to a discrete codebook value, $c_j \in C$:

$$z_q = q_1, q_2, ..., q_L, \text{ where } q_i = \min_{c_j \in C} \|e_i - c_j\| \tag{3}$$

The gradients are carried over from the decoder to the encoder by using a straight-through gradient estimator (Bengio, 2013) since this is a non-differentiable operation. The process is trained by using the loss function $L_{VQ}$

$$L_{VQ} = L_{rec} + \|sg[E(x)] - z_q\|_2^2 + \beta \|sg[z_q] - E(x)\|_2^2 \tag{4}$$

Where $L_{rec}$ stands for perceptual reconstruction loss (Esser et al., 2021; Zhang et al., 2018), $sg$ stands for stop gradient, $E(x)$ for the encoder values, and $z_q$ is the quantized latent. The generated latents are passed through the decoder to generate input samples $\hat{x} = G(z_q)$.

### 3.2 Discrete Diffusion Model

In the unleashing transformer paper (Bond-Taylor et al., 2022), at each forward time step $t$, the values of the latent are either kept the same or masked out entirely with probability $\frac{1}{t}$ and the reverse process gradually unveils the values. The reverse process predicts $p_\theta(z_0|z_t)$, reducing the stochastic nature of the training (Ho et al., 2020). The model was trained on the re-weighted ELBO that mimics the parameterized ELBO of continuous diffusion models.

$$\mathbb{E}_{q(z_0)} \left[ \sum_{t=1}^{T} \frac{T - t + 1}{T} \mathbb{E}_{q(z_t|z_0)} \left[ \sum_{[z_t]_i = m} \log p_\theta([z_0]_i|z_t) \right] \right] \tag{5}$$

Our approach adopts the same framework as Bond-Taylor et al. (2022). However, we adopt the Gaussian Blur as the forward diffusion process and directly predict $p_\theta(z_0|z_T)$. This enables us to create a single sampling step algorithm. In the following section, we will discuss how we adopted IHDM (Rissanen et al., 2022) and subsequently make a single-step sampling algorithm.

### 3.3 Inverse Heat Dissipation Model

In the second stage, we are training the latents obtained from the Vector-Quantized image model. IHDM applies Gaussian blur to a given data sample. This allows color entanglement and smooth interpolation of images (Rissanen et al., 2022). We adopt this forward diffusion process as part of our two-stage process. The difference between our approach and Rissanen et al. (2022) is how we utilize the Gaussian blur. Instead of calculating the Gaussian blur for each timestep $t$, we calculate the value for the final timestep $T$ directly. Then we train our diffusion model to predict $p_\theta(z_0|z_T)$. By restructuring the forward process we can now generate samples on a single step. For sampling, a blurry prior is created and stored during the initialization process. So whenever a sample needs to be produced the model takes the blurry prior and predicts $p_\theta(z_0|z_T)$.

---

**Algorithm 1** Sampling algorithm

---

$x \sim p(x_T)$                                          $\triangleright$ Sample from blurry prior during initialization
$x \leftarrow noise * \sigma + x$
$x_0 \leftarrow \text{Transformer}(x, labels)$                  $\triangleright$ Pass $x$ to Transformer, where labels are T
Return $x_0$

---

## 4 Experiment

### 4.1 VQGAN training

We used the ADNI dataset (Jack Jr et al., 2008) to train our VQGAN model. We preprocessed the MRI into 176 by 176. We used a codebook of size 1024, a latent dimension of 121, and a batch size of 3. We trained the model for 575000 steps. We used an adam (Kingma & Ba, 2014) optimizer with a learning rate of 0.0001 (warmup for 30000 steps).

### 4.2 VQ-IHDM training

Initially, from the VQGAN training we have a latent shape 120,121 where 120 is the number of slices and 121 is the dimension of each slice. We have taken the middle 40 slices and used them to train our VQ-IHDM model. For training, we have created a blur schedule with a maximum value of 24 and a minimum value of 0.5 for 200 timesteps. This blur schedule returns us an array of 200 values in descending order where the initial value is 24 and the final value is 0.5. This enables us to calculate how much Gaussian blur to apply for each timestep. We have taken a single slice from our training slices to create a blurry prior for sampling. We use Adam optimizer (Kingma & Ba, 2014) with a learning rate of 0.0001 (warmup for 30000 steps) and trained the model for 75000 steps with a batch size of 20. We have used a PC with NVIDIA GTX 1660 TI super GPU and 32 GB RAM to train our model. A BERT-style transformer (Devlin et al., 2018) model was used to learn the reverse diffusion process where the number of embeddings is 512, with 8 heads and 24 layers.

## 5 Evaluation

We have compared our approach to several GAN models and the diffusion model from the unleashing transformer. We have used the 3D code from CCEGAN (Xing et al., 2021), WGAN (Weng, 2019), VAEGAN (Yu et al., 2019), AlphaWGAN (Kwon et al., 2019) and converted all the layers and operations to 2D for training on the dataset. We calculated Frechet inception distance (FID), Kernel inception distance (KID), and Inception score (ISC) scores using the torch-fidelity package (Obukhov et al., 2020). Our results are given in the table 1, and our generated samples are provided in the figures 2, 3, 4, 5, 6. Additionally, we provide a step-by-step comparison with our approach and Bond-Taylor et al. (2022) to demonstrate what a single sample looks like at different time steps in the table 2. Since Discrete VQ-IHDM generates the sample in a single step we have provided the sample for only the first step.

Table 1: FID, KID, and ISC scores

| Model | FID(↓) | KID(↓) | ISC(↑) |
|---|---|---|---|
| Absorbing Diffusion (Bond-Taylor et al., 2022) | **28.46** | **0.025** | 1.77 |
| VQ-IHDM (ours) | 38.57 | 0.036 | **1.84** |
| VAEGAN (Yu et al., 2019) | 281.48 | 0.44 | 1.01 |
| CCEGAN (Xing et al., 2021) | 340.94 | 0.5 | **1.84** |
| AlphaWGAN (Kwon et al., 2019) | 146.93 | 0.18 | 1.66 |
| WGAN (Weng, 2019) | 212.91 | 0.25 | 1.42 |

| Model | 1 | 10 | 50 | 100 | 200 | 256 |
|---|---|---|---|---|---|---|
| Absorbing Diffusion | | | | | | |
| Discrete VQ-IHDM (Ours) | | | | | | |

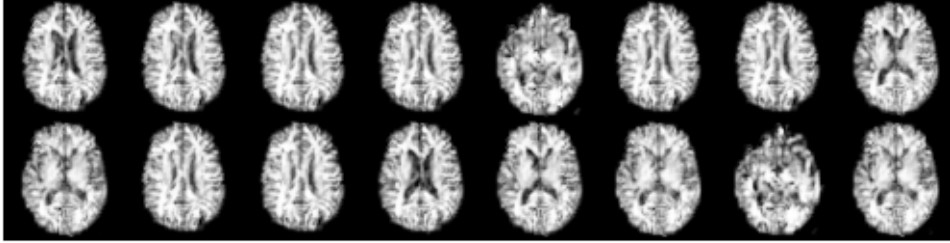

Table 2: Sinlge sample generation at different timestep.

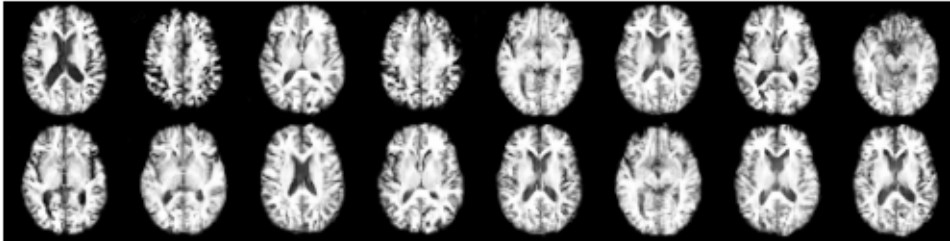

Figure 2: WGAN samples

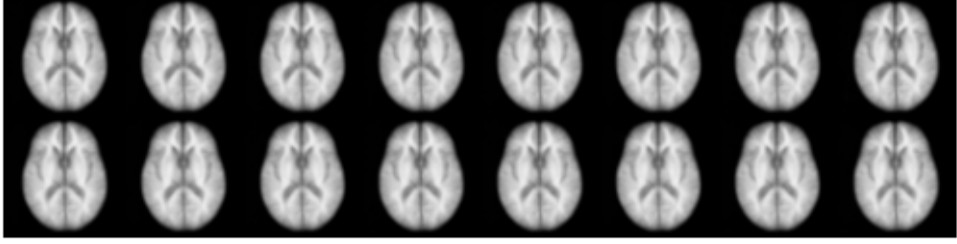

Figure 3: AlphaWGAN samples

Figure 4: VAEGAN samples

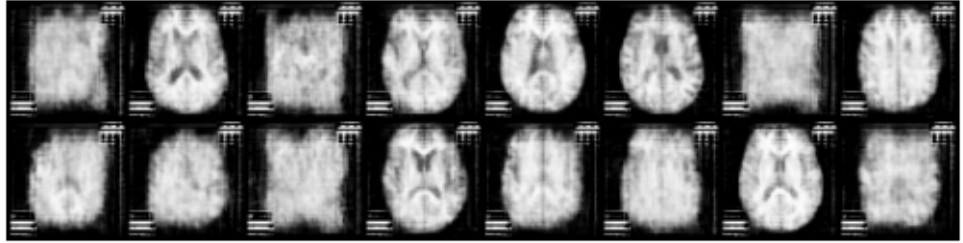

Figure 5: CCEGAN samples

## 6  Conclusion

We have created Discrete VQ-IHDM which is a hybrid generative model that combines VQGANs with IHDM. Our approach exploits the partial heat differential equation by using discrete cosine transformation, to directly calculate the output for the final timestep bypassing the Markov chain of forward steps in diffusion models. This allows us to learn directly from the final timestep to the initial state, and produce high-quality outputs. This work can be further extended by training on covariables such as age, sex, and subtype of diseases just like Pinaya et al. (2022) and by looking for mechanisms to generate higher resolution images. We hope that our model can serve the medical community in terms of producing synthetic datasets and creating educational content while preserving the privacy of patients.

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

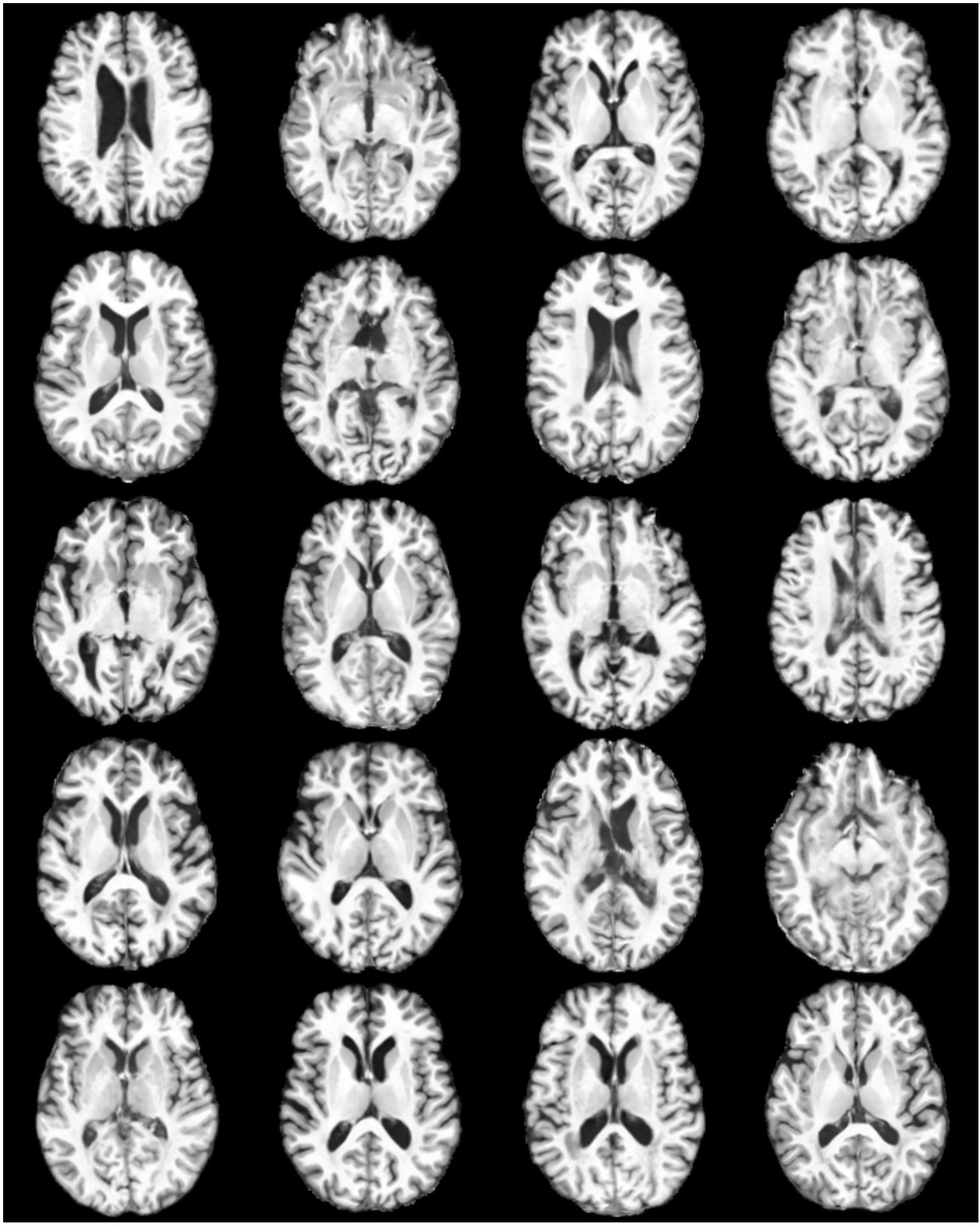

Figure 6: Samples from Discrete VQ-IHDM

Jonathan Ho, Ajay Jain, and Pieter Abbeel. Denoising diffusion probabilistic models. *Advances in Neural Information Processing Systems*, 33:6840–6851, 2020.

Jonathan Ho, Chitwan Saharia, William Chan, David J. Fleet, Mohammad Norouzi, and Tim Salimans. Cascaded diffusion models for high fidelity image generation, 2021.

Emiel Hoogeboom, Didrik Nielsen, Priyank Jaini, Patrick Forré, and Max Welling. Argmax flows and multinomial diffusion: Towards non-autoregressive language models. *arXiv preprint arXiv:2102.05379*, 3 (4):5, 2021.

Clifford R Jack Jr, Matt A Bernstein, Nick C Fox, Paul Thompson, Gene Alexander, Danielle Harvey, Bret Borowski, Paula J Britson, Jennifer L. Whitwell, Chadwick Ward, et al. The alzheimer's disease neuroimaging initiative (adni): Mri methods. *Journal of Magnetic Resonance Imaging: An Official Journal of the International Society for Magnetic Resonance in Medicine*, 27(4):685–691, 2008.

Tero Karras, Timo Aila, Samuli Laine, and Jaakko Lehtinen. Progressive growing of gans for improved quality, stability, and variation. *arXiv preprint arXiv:1710.10196*, 2017.

Amirhossein Kazerouni, Ehsan Khodapanah Aghdam, Moein Heidari, Reza Azad, Mohsen Fayyaz, Ilker Hacihaliloglu, and Dorit Merhof. Diffusion models for medical image analysis: A comprehensive survey. *arXiv preprint arXiv:2211.07804*, 2022.

Diederik P Kingma and Jimmy Ba. Adam: A method for stochastic optimization. *arXiv preprint arXiv:1412.6980*, 2014.

Gihyun Kwon, Chihye Han, and Dae-shik Kim. Generation of 3d brain mri using auto-encoding generative adversarial networks. In *Medical Image Computing and Computer Assisted Intervention–MICCAI 2019: 22nd International Conference, Shenzhen, China, October 13–17, 2019, Proceedings, Part III 22*, pp. 118–126. Springer, 2019.

Shao-Lun Lu, Heng-Chun Liao, Feng-Ming Hsu, Chun-Chih Liao, Feipei Lai, and Furen Xiao. The intracranial tumor segmentation challenge: Contour tumors on brain mri for radiosurgery. *Neuroimage*, 244: 118585, 2021.

Qing Lyu and Ge Wang. Conversion between ct and mri images using diffusion and score-matching models. *arXiv preprint arXiv:2209.12104*, 2022.

Puria Azadi Moghadam, Sanne Van Dalen, Karina C Martin, Jochen Lennerz, Stephen Yip, Hossein Farahani, and Ali Bashashati. A morphology focused diffusion probabilistic model for synthesis of histopathology images. In *Proceedings of the IEEE/CVF Winter Conference on Applications of Computer Vision*, pp. 2000–2009, 2023.

Alexander Quinn Nichol and Prafulla Dhariwal. Improved denoising diffusion probabilistic models. In *International Conference on Machine Learning*, pp. 8162–8171. PMLR, 2021.

Anton Obukhov, Maximilian Seitzer, Po-Wei Wu, Semen Zhydenko, Jonathan Kyl, and Elvis Yu-Jing Lin. High-fidelity performance metrics for generative models in pytorch. *Apache License*, 2020.

Muzaffer Özbey, Salman UH Dar, Hasan A Bedel, Onat Dalmaz, Şaban Özturk, Alper Güngör, and Tolga Çukur. Unsupervised medical image translation with adversarial diffusion models. *arXiv preprint arXiv:2207.08208*, 2022.

Cheng Peng, Pengfei Guo, S Kevin Zhou, Vishal M Patel, and Rama Chellappa. Towards performant and reliable undersampled mr reconstruction via diffusion model sampling. In *Medical Image Computing and Computer Assisted Intervention–MICCAI 2022: 25th International Conference, Singapore, September 18–22, 2022, Proceedings, Part VI*, pp. 623–633. Springer, 2022.

Walter HL Pinaya, Petru-Daniel Tudosiu, Jessica Dafflon, Pedro F Da Costa, Virginia Fernandez, Parashkev Nachev, Sebastien Ourselin, and M Jorge Cardoso. Brain imaging generation with latent diffusion models. In *Deep Generative Models: Second MICCAI Workshop, DGM4MICCAI 2022, Held in Conjunction with MICCAI 2022, Singapore, September 22, 2022, Proceedings*, pp. 117–126. Springer, 2022.

Severi Rissanen, Markus Heinonen, and Arno Solin. Generative modelling with inverse heat dissipation. *arXiv preprint arXiv:2206.13397*, 2022.

Robin Rombach, Andreas Blattmann, Dominik Lorenz, Patrick Esser, and Björn Ommer. High-resolution image synthesis with latent diffusion models, 2021.

Chitwan Saharia, Jonathan Ho, William Chan, Tim Salimans, David J. Fleet, and Mohammad Norouzi. Image super-resolution via iterative refinement, 2021.

Jascha Sohl-Dickstein, Eric A. Weiss, Niru Maheswaranathan, and Surya Ganguli. Deep unsupervised learning using nonequilibrium thermodynamics, 2015.

Jiaming Song, Chenlin Meng, and Stefano Ermon. Denoising diffusion implicit models. *arXiv preprint arXiv:2010.02502*, 2020.

Yang Song, Liyue Shen, Lei Xing, and Stefano Ermon. Solving inverse problems in medical imaging with score-based generative models, 2021.

Cathie Sudlow, John Gallacher, Naomi Allen, Valerie Beral, Paul Burton, John Danesh, Paul Downey, Paul Elliott, Jane Green, Martin Landray, et al. Uk biobank: an open access resource for identifying the causes of a wide range of complex diseases of middle and old age. *PLoS medicine*, 12(3):e1001779, 2015.

Abhishek Vahadane, Tingying Peng, Amit Sethi, Shadi Albarqouni, Lichao Wang, Maximilian Baust, Katja Steiger, Anna Melissa Schlitter, Irene Esposito, and Nassir Navab. Structure-preserving color normalization and sparse stain separation for histological images. *IEEE transactions on medical imaging*, 35(8):1962–1971, 2016.

Aaron Van Den Oord, Oriol Vinyals, et al. Neural discrete representation learning. *Advances in neural information processing systems*, 30, 2017.

Ashish Vaswani, Noam Shazeer, Niki Parmar, Jakob Uszkoreit, Llion Jones, Aidan N Gomez, Łukasz Kaiser, and Illia Polosukhin. Attention is all you need. *Advances in neural information processing systems*, 30, 2017.

Lilian Weng. From gan to wgan. *arXiv preprint arXiv:1904.08994*, 2019.

Shibo Xing, Harsh Sinha, and Seong Jae Hwang. Cycle consistent embedding of 3d brains with auto-encoding generative adversarial networks. In *Medical Imaging with Deep Learning*, 2021.

Xianwen Yu, Xiaoning Zhang, Yang Cao, and Min Xia. Vaegan: A collaborative filtering framework based on adversarial variational autoencoders. In *IJCAI*, pp. 4206–4212, 2019.

Richard Zhang, Phillip Isola, Alexei A Efros, Eli Shechtman, and Oliver Wang. The unreasonable effectiveness of deep features as a perceptual metric. In *Proceedings of the IEEE conference on computer vision and pattern recognition*, pp. 586–595, 2018.

