# OpenReview forum: "Discrete VQ-IHDM: MRI Generation with Vector Quantized Inverse Heat Dissipation Model"
_TMLR — Rejected by TMLR_

### Review · Reviewer_2kXu · 2023-12-25

**Summary Of Contributions:**

Authors simply replace a conventional diffusion model for sampling in the __extant__ Vector-Quantized Codes-based GAN with the inverse-heat-dissipation-model based diffusion model to generate MR images.

**Audience:**

No

**Claims And Evidence:**

No

**Requested Changes:**

Questions on specific statements made throughout the manuscript:

--> What is the sense of ‘color’ in the setting of this manuscript? Authors have repeatedly mentioned “the color entanglement” per the citation Rissanen et al., 2022. Why mention this?

--> Equation (5) is not ELBO. Authors must clearly and explicitly write down what the loss function is to optimize for training the diffusion model.

--> In Table 1, except for the metric of inception score, the proposed method VQ-IHDM is worse than other baseline methods.This clearly indicates that either the VQ-IHDM code authors use is very difficult to tune for a strong performance or more efforts are needed for improved finetuning.
The current numerical result cannot establish the validity of the proposed method.

--> Regarding establishment of technical correctness: there is completely no report on the training record of the vector-quantized GAN, so there is no evidence provided for reviewers to determine whether or not the GAN is properly trained.

--> The training of VQ-IHDM is partitioned into two steps: training of VQGAN to learn the structure of latent variable space, and training of IHDM. It is unclear what the ground truth of “deblurred latent code” and what noisy input are, respectively, for IHDM training.

--> Many technical details are missing, including what data pairs are used for training at each step, algorithm details, convergence proof, fine characterisation of generated dataset, and discussion of stability on hyperparameters.

**Strengths And Weaknesses:**

Authors fail to clearly explain the main purpose of this manuscript. If the key point the generation of a useful dataset for the medical image community, then a thorough description of the generated dataset should be presented. Simply presenting a method of generating synthetic dataset without actually presenting the final dataset is largely invalid in establishing the reliability or effectiveness of the proposed method.

There is essentially no novel or new methodology proposed in this work, and the provided numerical evidence cannot support the scientific correctness of the proposed approach.

---

### Review · Reviewer_peuZ · 2024-01-23

**Summary Of Contributions:**

In this paper, the authors propose a new method for automagical MRI image generation. Their method combines two existing methods by using VQGAN to create discrete code representation of training images first, then employing IHDM for the reverse process of image generation. Experimental results showed that the proposed has high image fidelity and comparable performance as the Absorbing Diffusion method.

**Audience:**

Yes

**Claims And Evidence:**

No

**Requested Changes:**

1. Many mathematical symbols are used in the paper without definitions. It’s recommended to go through the paper and double check all symbols used.
2. Code segment in page 2 is not very informative and could be removed.
3. Eq. (4), can you define the perceptual reconstruction loss here in the paper? I understand the authors listed referring doc of the loss. It would be convenient for the readers to read them directly from this paper.
4. In the experiment section, it’s recommended  to show a diagram showing sizes of images/preprocessed images/latent variables at  each step. That will help the audience to understand the whole process quickly.
5. The authors are encouraged to analyze why the proposed method has larger FID and KID compared to the Absorbing Diffusion method.

**Strengths And Weaknesses:**

Strengths:
1. Applying VQGAN to compress the input image and generate them using inverse heat dissipation based model sound novel. If the author can further link the inverse heat dissipation process with the generation process of MRI image, it may support this work from the view point of physical process and make more impact to this field. For example, in the imaging process of the MRI image, is there heat dissipation process involved (e.g. blood flow, contrast agent) which will affect the imaging quality?
2. Experimental results look promising with comparable performance to the SOTA image generation methods.

Weaknesses:
1. My major concern is that methodology novelty/contribution of this work is limited. Basically it’s combined of two existing method with minor modifications.
2. Motivation of the method is not well clarified. For example, why should we use VQGAN to generate discrete code? Compression usually comes with cost on information loss. For the dilemma of image quality (generated from the proposed method) and computational  cost, personally I prefer image quality in the medical area.
3. Figure 1 looks confusing. I have hard time to understand where are the two components VQGAN and IHDM in the diagram and how they interact with each other. Are inputs of IHDM the code slices learned from the VQGAN? Is Gaussian blur applied to the compressed code slices or the original images?
4. Experimental section looks not solid enough to justify publication at TMLR.

---

### Review · Reviewer_w74G · 2024-01-29

**Summary Of Contributions:**

This paper works on 2D MRI image generation using diffusion models. The proposed method first employs VQGAN to compress the images into latent codes and then trains a diffusion model in the latent space with Gaussian blur as the corruption method.

**Audience:**

Yes

**Broader Impact Concerns:**

Since this paper works on the generation of medical data, a discussion is needed regarding aspects like privacy, safety, and other ethical issues.

**Claims And Evidence:**

No

**Requested Changes:**

More insights and evaluation may add value. But this may go beyond the scope of this submission cycle.

**Strengths And Weaknesses:**

Strengths:
1. This paper is well-written and easy to follow.
2. This paper works on diffusion models for MRI generation, which is an interesting and meaningful topic.

Weaknesses:
1. The proposed method combines several existing methods, such as VQGAN and IHDM. A direct application and combination of existing models to the MRI domain can be fine, but more insights are needed. What is the motivation and benefit of these models for MRI particularly? This is not clearly stated either by intuition or experiments.
2. In section 4.2, it is stated that "we have taken a single slice from our training slices to create a blurry prior for sampling". Do we need a different blurry prior for each training subject?
3. The evaluation is not convincing enough. In Table 1, the proposed VQ-IHDM does not outperform Absorbing Diffusion. Besides, no ablation study validates each design component in the proposed method.

---

### Decision · Action_Editor_nuTr · 2024-04-02

**Recommendation:** Reject

**Comment:**

As the reviewers have pointed out there is a lack of experimental evaluation and detailed discussion in the paper that makes it unsuitable for publishing as a method for MRI generation. Furthermore, the authors did not provide a rebuttal.

**Audience:**

As mentioned by reviewer w74G, a direct application and combination of existing models to the MRI domain is publishable. Still, the paper lacks motivation and does not provide any insights as to why this method should be used instead of methods already published in the literature. It is unlikely the paper will find interest in the applied MRI audience in TMLR.

**Claims And Evidence:**

MRI generation is a vast field in medical imaging. The method proposed is a straightforward combination of previously proposed approaches and seems to provide some partial solution to the high-fidelity MRI generation problem in a significantly reduced set of experiments provided in the paper. Given the extent of the research in the field, the claims of high-fidelity MRI generation have been poorly supported and require a more extensive experimental evaluation.